# 4DLANGVGGT: 4D LANGUAGE-VISUAL GEOMETRY GROUNDED TRANSFORMER

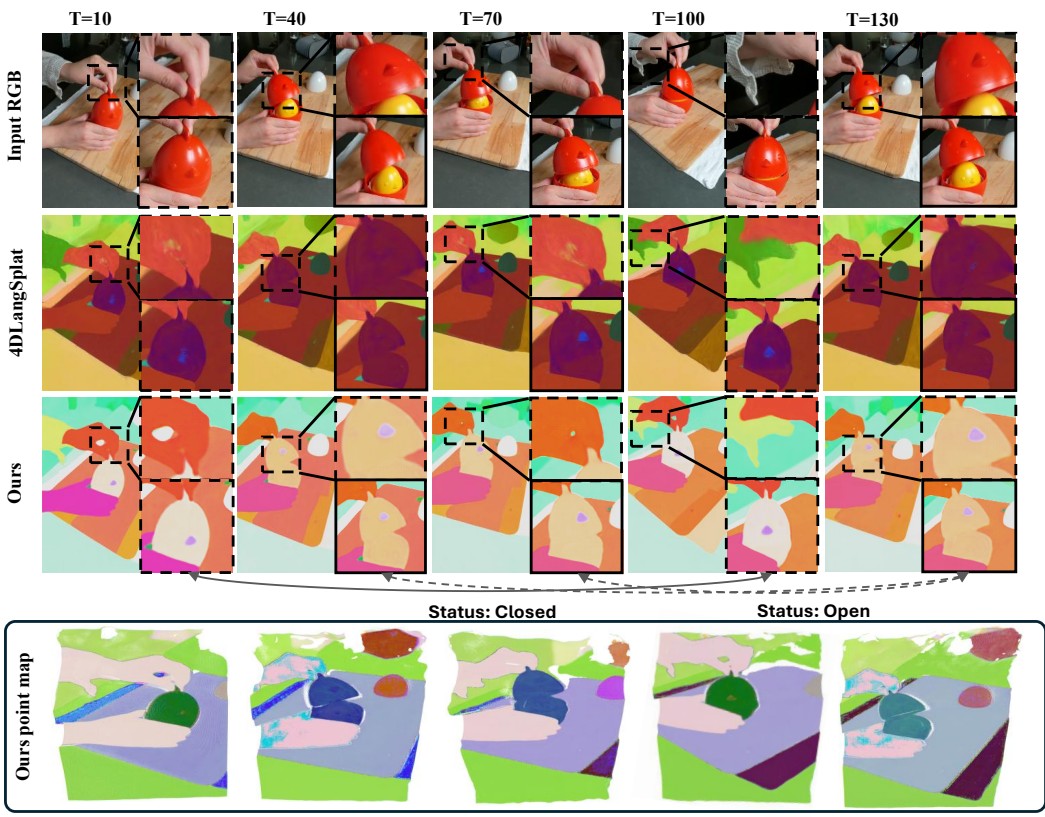

Figure 1: Qualitative comparison of language feature visualizations by our method and 4DLangSplat (Li et al., 2025c). The top part shows semantic visualizations learned by both methods, where our approach not only captures finer details (upper-right zoomed regions) but also demonstrates higher sensitivity to temporal state changes of objects (lower-right highlighted examples). The bottom part illustrates our method's learned semantic features projected onto 3D point clouds, providing a more interpretable view of spatiotemporal semantics in dynamic scenes.

## ABSTRACT

Constructing 4D language fields is crucial for embodied AI, augmented/virtual reality, and 4D scene understanding, as they provide enriched semantic representations of dynamic environments and enable open-vocabulary querying in complex scenarios. However, existing approaches to 4D semantic field construction primarily rely on scene-specific Gaussian splatting, which requires per-scene optimization, exhibits limited generalization, and is difficult to scale to real-world applications. To address these limitations, we propose **4DLangVGGT**, the first Transformer-based feed-forward unified framework for 4D language grounding, that jointly integrates geometric perception and language alignment within a single architecture. 4DLangVGGT has two key components: the 4D Visual Geometry Transformer, StreamVGGT, which captures spatio-temporal geometric representations of dynamic scenes; and the Semantic Bridging Decoder (SBD), which projects geometry-aware features into a language-aligned semantic space, thereby

enhancing semantic interpretability while preserving structural fidelity. Unlike prior methods that depend on costly per-scene optimization, 4DLangVGGT can be jointly trained across multiple dynamic scenes and directly applied during inference, achieving both efficiency and strong generalization. This design significantly improves the practicality of large-scale deployment and establishes a new paradigm for open-vocabulary 4D scene understanding. Experiments on HyperNeRF and Neu3D datasets demonstrate that our approach not only generalizes effectively but also achieves state-of-the-art performance, achieving up to **2%** gains under per-scene training and **1%** improvements under multi-scene training. Our code released in 4DLangVGGT Repository.

# 1 INTRODUCTION

Scene understanding (Peng et al., 2023) has become a core capability in modern applications such as human–robot interaction (Fang et al., 2024), AR/VR content creation (Schieber et al., 2025), and intelligent surveillance (Yuan et al., 2024). While recent advances in 3D visual–language learning (Ma et al., 2025; Fan et al., 2025) have shown strong performance in static settings, they remain limited when extended to dynamic 4D scenarios, where both geometry and semantics evolve continuously over time. Unlike static environments (Li et al., 2025a;b; Qin et al., 2024), real-world scenes demand temporal consistency, semantic continuity, and cross-frame alignment to handle open-ended and time-sensitive queries. Directly applying 3D methods often leads to semantic drift and unstable alignment, highlighting a critical gap that motivates research in robust 4D vision–language models (Cai et al., 2025; Ge et al., 2025).

Recent research (Li et al., 2025c) has begun to explore extending scene representations toward language-guided 4D fields. However, most existing approaches remain heavily reliant on Gaussian Splatting pipelines. While Gaussian Splatting has shown promising performance in controlled settings, its fundamental drawback lies in the need for explicit per-scene optimization. This requirement introduces several critical limitations: the computational cost becomes prohibitively high, scalability across diverse videos is severely restricted, and separate models must be maintained for different environments, making large-scale deployment impractical. More importantly, the reliance on per-scene training fundamentally undermines the feasibility of real-time applications, where efficiency and generalization are indispensable requirements. These constraints underscore the pressing need for new solutions that move beyond scene-specific pipelines.

To alleviate the scalability issues caused by per-scene optimization, we turn to the paradigm of feedforward 4D geometric reconstruction (Zhuo et al., 2025; Wang et al., 2024; 2025b). Methods such as StreamVGGT (Zhuo et al., 2025) demonstrate strong real-time performance and generalization by enabling efficient reconstruction without scene-specific optimization. However, these approaches focus solely on geometry and motion, lacking semantic or language alignment, and are therefore insufficient for supporting open-vocabulary 4D understanding. This gap highlights the need for a next-generation framework that jointly models geometry and semantics within a unified architecture.

To address these limitations, we propose **4DLangVGGT**, a Transformer-based feed-forward framework that unifies dynamic geometric reconstruction and visual-language alignment within a single architecture. The framework integrates two key components: a 4D Visual Geometry Transformer, which captures spatio-temporal geometric representations of dynamic scenes, and a Semantic Bridging Decoder (SBD), which maps scene-aware features into a language-aligned semantic space to bridge the gap between geometric perception and semantic prediction. Through this design, the model achieves both high structural fidelity and semantic consistency, as shown in Fig. 1, while inheriting the efficiency and strong generalization capabilities of feed-forward approaches. More importantly, to the best of our knowledge, our proposed 4DLangVGGT is *the first unified language field model* that can be jointly trained across multiple dynamic scenes and directly applied during inference, eliminating the need for costly per-scene optimization and thereby significantly enhancing the practicality of deployment in large-scale, real-world systems. Experiments show that our method not only generalizes well but also achieves state-of-the-art results across multiple benchmarks, yielding up to **2%** improvements under per-scene training and around **1%** gains under training across scenes.

Our main contributions are as follows:

- We propose 4DLangVGGT, the first Transformer-based feed-forward framework that unifies 4D geometric reconstruction with visual-language alignment in a single network.

- We introduce SBD, which maps dynamic, scene-aware features into a language-aligned semantic space, effectively bridging the gap between geometric perception and semantic prediction.

- Unlike prior scene-specific methods, our model can be jointly trained across multiple dynamic scenes (6 scenes in HyperNeRF and 6 scenes in Neu3D) and directly applied at inference without per-scene optimization, making large-scale real-world deployment feasible.

## 2 RELATED WORKS

**Static 3D Scene Understanding.** Language grounding in 3D has been studied with NeRF-based and Gaussian-based representations. NeRF-based methods such as LERF (Kerr et al., 2023) and OV-NeRF (Liao et al., 2024) enabled open-vocabulary querying but suffered from slow volumetric rendering. To improve efficiency, LangSplat (Qin et al., 2024) adopted 3D Gaussian Splatting (Kerbl et al., 2023) with hierarchical semantics, achieving orders-of-magnitude faster rendering, while extensions like GaussianGrasper (Zheng et al., 2024) demonstrated applications in robotics. Multimodal fusion approaches such as LangSurf (Li et al., 2024) further enhanced cross-modal alignment. Nonetheless, existing methods remain limited to static scenes and do not generalize to dynamic environments.

**Dynamic 4D Scene Understanding.** Bridging natural language and dynamic 4D scene understanding has emerged as a critical research direction, with core efforts focused on tight integration of linguistic semantics into time-varying geometric representations. *4DLangSplat* (Li et al., 2025c) uses object-wise video captions and a status deformable network to supervise a 4D Gaussian Splatting field that supports both time-sensitive and time-agnostic open-vocabulary queries; *4-LEGS* (Fiebelman et al., 2025) lifts spatio-temporal video features into a 4D Gaussian representation to localize text prompts in space and time, allowing interactive video editing. They both depend on Gaussian Splatting, which needs scene-specific optimization. Collectively, these works represent important advances but do not yet satisfy all desiderata of efficient inference, cross-scene generalization, and tightly aligned semantics with evolving geometry.

**Feed-forward Scene Reconstruction.** Feed-forward frameworks provide a scalable alternative to NeRF- and GS-based reconstruction by leveraging pretrained encoders or end-to-end architectures. Works such as DUST3R (Wang et al., 2024), VGGT (Wang et al., 2025a), and StreamVGGT (Zhuo et al., 2025) enable efficient 3D and 4D reconstruction, while methods like SplatterImage (Szymanowicz et al., 2024), Flash3D (Szymanowicz et al., 2025), and Niagara (Wu et al., 2025) emphasize efficiency and scalability. However, these approaches focus solely on geometric reconstruction, leaving open the challenge of unifying feed-forward reconstruction with language grounding for generalizable 4D semantic understanding.

## 3 PRELIMINARIES: VGGT & STREAMVGGT

Visual Geometry Grounded Transformer (VGGT) (Wang et al., 2025a) is a feed-forward Transformer for 3D scene reconstruction that achieves fast and accurate results in a single pass. Given one or more scene views, it directly predicts key 3D attributes such as camera parameters, depth maps, point maps, and 3D point tracks. The processing flow of VGGT can be summarized in three stages. First, the image encoder DINO (Caron et al., 2021; Oquab et al., 2023), denoted as $\mathcal{E}$, transforms the input sequence $\{I_t\}_{t=1}^{T}$ into image tokens $\{F_t\}_{t=1}^{T}$, with an additional camera token $\{C_t\}_{t=1}^{T}$ appended to each image. These tokens are then fed into the Alternating-Attention transformer layers $\mathcal{D}$, which alternate between frame-level and cross-frame self-attention to refine the representations and produce two outputs: updated camera tokens and geometry tokens $\{G_t\}_{t=1}^{T}$. Finally, the multi-head predictor, comprising the camera head $\mathcal{H}_{\text{cam}}$ and the DPT (Ranftl et al., 2021) head $\mathcal{H}_{\text{DPT}}$, decodes the corresponding tokens to yield camera parameters $O_t^c$ and dense geometric predictions $O_t^g$, thereby completing the end-to-end mapping from images to 3D attributes.

StreamVGGT extends VGGT to the streaming setting by employing causal temporal attention for sequential inference, where each incoming frame is processed incrementally. During inference, only a cache memory of past tokens needs to be maintained: $\mathbb{M}_t = \mathbb{M}_{t-1} \cup [C_t, F_t]$, enabling real-time

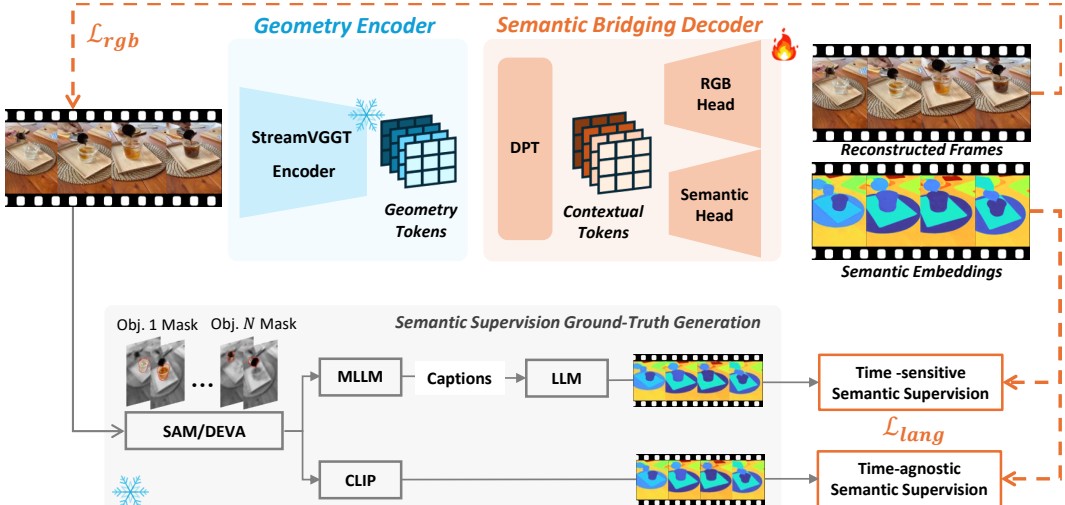

Figure 2: Overview of 4DLangVGGT. The framework integrates a geometry encoder, a semantic bridging decoder, and a multi-objective training strategy to achieve language-aware 4D fields with geometric fidelity and semantic alignment.

efficiency while preserving temporal consistency. The overall process can be formulated as follows:

$$\boldsymbol{F}_t = \mathcal{E}(\boldsymbol{I}_t); \quad [\boldsymbol{C}_t, \boldsymbol{G}_t] = \mathcal{D}\left(\,[\boldsymbol{C}_t, \boldsymbol{F}_t]\mid \mathbb{M}_{t-1}\,\right); \quad \boldsymbol{O}_t^c = \mathcal{H}_{\text{cam}}(\boldsymbol{C}_t),\ \boldsymbol{O}_t^g = \mathcal{H}_{\text{DPT}}(\boldsymbol{G}_t). \quad (1)$$

These definitions and formulations provide the necessary background for introducing our method.

## 4 METHODOLOGY: 4DLANGVGGT

We introduce **4DLangVGGT**, a unified framework for building *language-aware 4D fields* that maintain geometric fidelity while ensuring semantic alignment. As illustrated in Figure 2, the framework comprises three main components: (i) a StreamVGGT-based geometry encoder that generates spatio-temporal geometric representations (Sec. 4.1), (ii) a Semantic Bridging Decoder (SBD) that maps geometry tokens into a language-aligned semantic space (Sec. 4.2), and (iii) a multi-objective training strategy that jointly optimizes semantic alignment and appearance reconstruction (Sec. 4.3). Together, these components provide a robust foundation for 4D perception that is both structurally faithful and semantically interpretable.

### 4.1 STREAMVGGT-BASED GEOMETRY ENCODER

As mentioned in the preliminaries and Eq. (1), the StreamVGGT aggregator alternates between *spatial attention* and *causal temporal attention*, producing geometry tokens $\{\boldsymbol{G}_t\}_{t=1}^{T}$ that encode both fine-grained 3D geometry structure and temporal dynamics. In our framework, we adopt this architecture but keep it frozen during training. The reason is twofold: (i) StreamVGGT has already been pre-trained on large-scale video data for geometry reconstruction, providing strong spatio-temporal representations that generalize well to diverse scenes; and (ii) freezing this part avoids redundant optimization and reduces computational cost, allowing the training process to focus on semantic alignment rather than relearning geometry from scratch.

In our framework, we leverage both geometry tokens and camera tokens. The geometry tokens $\boldsymbol{G}_t$ ensure geometry-centered representations that serve as the foundation for semantic alignment. Meanwhile, the camera tokens $\boldsymbol{C}_t$ are retained mainly for inference. They remain frozen during training, but at inference time they enable the model to exploit camera intrinsics and extrinsics to map features back into the 4D point cloud space, ensuring that semantic information is properly injected and aligned at the point cloud level. The StreamVGGT-based encoder provides a strong, geometry-centered foundation for our framework, enabling reliable spatio-temporal representations to support subsequent semantic alignment.

## 4.2 SEMANTIC BRIDGING DECODER (SBD)

While geometry tokens capture the geometry structural and temporal dynamics of a scene, they remain agnostic to semantics and cannot directly align with natural language queries. To address this gap, we propose the Semantic Bridging Decoder (SBD), whose goal is to establish a robust mapping between geometry representations and language semantics, thereby unifying geometric fidelity and semantic alignment.

**Geometry-to-Contextual Representation Transformation.** The input geometry tokens $G_t t = 1^T$ are first processed by a contextual-aware Dense Prediction Transformer (DPT) (Ranftl et al., 2021). DPT combines the spatial sensitivity of local convolutional operations with the global modeling capability of Transformers, thereby capturing long-range dependencies across both spatial and temporal dimensions. This new introduced DPT, denoted as $\mathcal{H}_{\text{DPT}}^{\text{lang}}$, which employs stacked self-attention layers to transform geometry tokens into contextually enriched feature representations, significantly enhancing their semantic discriminability.

$$H_t = \mathcal{H}_{\text{DPT}}^{\text{lang}}(G_t), \quad \forall t \in [1, \cdots, T], \tag{2}$$

where $H_t \in \mathbb{R}^{h \times w \times c}$ is referred to as the unified 4D feature representation. Here, $h$ and $w$ denote the spatial resolution of the token map, while $c$ is the feature dimension after transformation. Importantly, this module remains trainable during optimization, allowing it to be continuously refined for semantic tasks.

**Dual-head Semantic and Reconstruction Decoding.** Once the contextual-geometry features $H_t$ are obtained, they are passed through two independent prediction heads that project them into complementary semantic and visual subspaces for dual supervision:

$$\hat{S}_t = f_{\text{Lang}}(H_t), \quad \hat{I}_t = \sigma(f_{\text{RGB}}(H_t)), \quad \forall t \in [1, \cdots, T], \tag{3}$$

where $f_{\text{Lang}}$ maps the features into a $d$-dimensional semantic embedding space for language alignment, yielding $\hat{S}_t \in R^{h \times w \times d}$, which serves as the predicted semantic representation at time $t$. Meanwhile, $f_{\text{RGB}}$ projects the features back into the image space to reconstruct RGB frames, producing $\hat{I}_t \in R^{H \times W \times 3}$, which represents the reconstructed videos at time $t$ and thereby enforces perceptual consistency. Here, $H$ and $W$ denote the spatial resolution of video frames.

## 4.3 MULTI-OBJECTIVE TRAINING

**Semantic Loss.** During training, we employ two complementary types of semantic supervision: time-agnostic semantic supervision and time-sensitive semantic supervision. The former provides static, object-level constraints, while the latter captures temporally evolving semantics, and together they enhance the model's ability to achieve robust semantic alignment. For each video, we first use Segment Anything Model (SAM) (Kirillov et al., 2023) and DEVA (Cheng et al., 2023) to generate its object-level masks $\{M_{i,t}\}_{i,t=1,1}^{N,T}$, where $i$ denotes the object index with $N$ objects in total.

*Time-agnostic semantic supervision.* Each mask region is passed through CLIP (Radford et al., 2021a) to obtain its object-specific embedding, which is then assigned to all pixels within the mask region, yielding a region-aligned semantic feature map:

$$e_{i,t}^{\text{CLIP}} = f_{\text{CLIP}}(I_t \cdot M_{i,t}), \quad S_t^{\text{CLIP}} = \sum_{i=1}^{N} e_{i,t}^{\text{CLIP}} \cdot M_{i,t}, \quad \forall t \in [1, \cdots, T] \tag{4}$$

where the CLIP embedding is denoted as $e_{i,t}^{\text{CLIP}} \in R^{1 \times 1 \times d}$, and the object mask $M_{i,t} \in R^{h \times w \times 1}$ takes the value 1 if a pixel belongs to the object and 0 otherwise.

*Time-sensitive semantic supervision.* Using SAM masks across frames, we feed the video-level regions corresponding to each object into a multimodal large language model ($f_{\text{MLLM}}$) to generate detailed and temporally consistent descriptions. These descriptions are then encoded by a large language model ($f_{\text{LLM}}$) to obtain the corresponding semantic embeddings, which will be assigned to all pixels within the mask, producing dynamic semantic ground truth:

$$\{e_{i,t}^{\text{dyn}}\}_{t=1}^{T} = f_{\text{LLM}}\left(f_{\text{MLLM}}(\{I_t \cdot M_{i,t}\}_{t=1}^{T})\right), \quad S_t^{\text{dyn}} = \sum_{i=1}^{N} e_{i,t}^{\text{dyn}} \cdot M_{i,t}. \tag{5}$$

*Final semantic supervision.* The semantic maps $\hat{S}_t$ predicted by the Semantic Head in Eq. (3) are aligned with ground truth $S_t \in \{S_t^{\text{CLIP}}, S_t^{\text{dyn}}\}$ using a combination of $\mathcal{L}_1$ regression and cosine

similarity.

$$\mathcal{L}_{\text{lang}} = \sum_{t=1}^{T} \lambda_1 |\hat{\boldsymbol{S}}_t - \boldsymbol{S}_t|_1 + \lambda_2 \left(1 - \cos(\hat{\boldsymbol{S}}_t, \boldsymbol{S}_t)\right), \quad \forall \boldsymbol{S}_t \in \{\boldsymbol{S}_t^{\text{CLIP}}, \boldsymbol{S}_t^{\text{dyn}}\}. \tag{6}$$

Here, $\lambda_1$ and $\lambda_2$ are loss weights. This dual supervision scheme enables the model to learn both static object-level semantics and temporally dynamic semantics, thereby improving alignment in dynamic scenes.

**Reconstruction Loss.** To ensure perceptual fidelity, the reconstructed RGB frames are supervised using a hybrid $\mathcal{L}_1$–$\mathcal{L}_2$ objective:

$$\mathcal{L}_{\text{rgb}} = \sum_{t=1}^{T} \lambda_{\text{img}} \|\hat{\boldsymbol{I}}_t - \boldsymbol{I}_t\|_1 + (1 - \lambda_{\text{img}}) \|\hat{\boldsymbol{I}}_t - \boldsymbol{I}_t\|_2^2, \tag{7}$$

where $\hat{\boldsymbol{I}}_t$ is the frame reconstructed by the RGB Head in Eq. (3), $\boldsymbol{I}_t$ is the ground-truth input frame. $\lambda_{\text{img}} \in [0, 1]$ controls the trade-off between the structural accuracy ($\mathcal{L}_1$) and the pixel-level smoothness ($\mathcal{L}_2$).

**Final Joint Objective.** To jointly preserve semantic alignment and visual fidelity, we employ a dual-supervision scheme. The overall training objective is defined as

$$\mathcal{L} = \alpha \mathcal{L}_{\text{lang}} + \beta \mathcal{L}_{\text{rgb}}, \tag{8}$$

where $\alpha, \beta \geq 0$ control their relative contributions.

## 5 EXPERIMENTS

### 5.1 EXPERIMENTAL SETUP

**Training Data.** We conducted training and evaluation on the HyperNeRF (Park et al., 2021) and Neu3D (Li et al., 2022) datasets. We utilized the semantic segmentation annotation dataset for dynamic scenes provided by 4DLangSplat Li et al. (2025c). For feature extraction, the OpenCLIP ViT-B/16 model was used to obtain CLIP features, while the Qwen2.5-VL-7B-Instruct Bai et al. (2025) model was employed to extract dynamic semantics. Following 4DLangSplat, the e5-mistral-7b model was applied to process time-varying captions and generate embeddings, and separate autoencoders were trained to compress the CLIP Radford et al. (2021b) features to 3 dimensions and the dynamic semantics features to 6 dimensions.

**Implementation Details.** The aggregator module of StreamVGGT (Zhuo et al., 2025) was used to extract geometric features from input video sequences, with a maximum of 128 past frames retained to preserve temporal dependencies while controlling memory usage. Following StreamVGGT (Zhuo et al., 2025) and VGGT (Wang et al., 2025a), input frames were resized to 518 pixels; however, we instead cropped them to the nearest multiple of 14 to better approximate the original resolution. Training employed a batch size of 8 and an initial learning rate of $4 \times 10^{-5}$, and all experiments were conducted on four NVIDIA GeForce RTX 3090 GPUs (24 GB).

**Baselines.** Following the evaluation protocol of 4DLangSplat, we benchmark our approach against representative methods under both time-agnostic and time-sensitive query settings. Our primary baselines are LangSplat (Qin et al., 2024), which introduces language-driven Gaussian splatting for static scene understanding, and 4DLangSplat (Li et al., 2025c), which extends this paradigm to dynamic scenes by incorporating temporal modeling. In the time-agnostic setting, we further compare against Feature-3DGS (Zhou et al., 2024), a feature distillation framework that compresses high-dimensional representations into compact 3D Gaussians, and Gaussian Grouping (Ye et al., 2024), which leverages semantic segmentation to cluster and render scene elements. For the time-sensitive setting, we include the deformable CLIP from 4DLangSplat, which integrates deformable Gaussian fields with static CLIP embeddings to assess cross-modal alignment, and Non-Status Field, which removes temporal state modeling to isolate its contribution.

The definitions of the evaluation metrics, together with additional implementation details and analysis, are provided in the appendix for completeness.

Table 1: Quantitative comparison of **time-agnostic language queries** on the HyperNeRF dataset.

| Method | Per-scene | americano | | chick-chicken | | split-cookie | | torchocolate | | Average | |
|---|---|---|---|---|---|---|---|---|---|---|---|
| | | mIoU | mAcc | mIoU | mAcc | mIoU | mAcc | mIoU | mAcc | mIoU | mAcc |
| LangSplat (Qin et al., 2024) | ✓ | 72.08 | 97.61 | 75.98 | 97.86 | 76.54 | 97.32 | 69.55 | 98.09 | 73.54 | 97.72 |
| Feature-3DGS (Zhou et al., 2024) | ✓ | 34.65 | 62.96 | 47.21 | 87.22 | 47.03 | 68.25 | 24.71 | 64.58 | 38.40 | 70.75 |
| Gaussian Grouping (Ye et al., 2024) | ✓ | 61.77 | 71.31 | 34.65 | 75.52 | 72.71 | 96.56 | 58.95 | 85.52 | 57.02 | 82.22 |
| 4DLangSplat (Li et al., 2025c) | ✓ | 83.48 | 98.77 | 86.50 | 98.81 | 90.04 | 98.67 | 71.79 | 98.10 | 82.95 | 98.59 |
| 4DLangVGGT (Ours) | ✓ | **86.45** | **98.95** | **90.70** | **99.03** | **90.15** | **98.79** | **72.77** | **98.32** | **85.02** | **98.77** |
| 4DLangVGGT (Ours) | ✗ | 82.46 | 98.36 | 86.91 | 98.88 | 91.44 | 98.87 | 75.15 | 98.57 | 83.99 | 98.67 |

Table 2: Quantitative comparison of **time-sensitive language queries** on the HyperNeRF dataset.

| Method | Per-scene | americano | | chick-chicken | | split-cookie | | espresso | | Average | |
|---|---|---|---|---|---|---|---|---|---|---|---|
| | | Acc | vIoU | Acc | vIoU | Acc | vIoU | Acc | vIoU | Acc | vIoU |
| LangSplat (Qin et al., 2024) | ✓ | 45.19 | 23.16 | 53.26 | 18.20 | 73.58 | 33.08 | 44.03 | 16.15 | 54.01 | 22.65 |
| Deformable CLIP (Li et al., 2025c) | ✓ | 60.57 | 39.96 | 52.17 | 42.77 | 89.62 | 75.28 | 44.85 | 20.86 | 61.80 | 44.72 |
| Non-Status Field (Li et al., 2025c) | ✓ | 83.65 | 59.59 | 94.56 | 86.28 | 91.50 | 78.46 | 78.60 | 47.95 | 87.58 | 68.57 |
| 4DLangSplat (Li et al., 2025c) | ✓ | 89.42 | 66.07 | 96.73 | 90.62 | 95.28 | 83.14 | 81.89 | 49.20 | 90.83 | 72.26 |
| 4DLangVGGT (Ours) | ✓ | **89.96** | **66.78** | **98.01** | **93.56** | **95.56** | **83.44** | **82.44** | **51.56** | **90.86** | **73.06** |
| 4DLangVGGT (Ours) | ✗ | 90.03 | 67.77 | 97.81 | 93.44 | 95.76 | 84.02 | 82.86 | 52.06 | 91.44 | 74.74 |

## 5.2 MAIN RESULTS

We evaluate two training regimes to examine both cross-scene applicability and per-scene performance. The first regime trains a single model on multiple videos and applies this shared model for inference across different scenes ("multi-video single model"). The second regime adopts the per-scene protocol used in 4DLangSplat, i.e., training one model per scene. This per-scene setting is included to align with existing Gaussian splatting methods and to provide a fair comparison of our method's performance.

### 5.2.1 HYPERNERF DATASET

We evaluate on HyperNeRF under two modes: time-agnostic and time-sensitive language queries, assessing both spatial grounding accuracy and temporal dynamics.

**Time-Agnostic Language Queries.** As shown in Table 1, under the per-scene setting (training one model per scene with the same protocol as 4DLangSplat), training and testing sets are not fully disjoint, and the results reflect performance under the same distribution. In this setting, 4DLangVGGT consistently surpasses all baselines, outperforming 4DLangSplat by 3% mIoU and 0.18% mAcc on average. Under the multi-video single-model setting (a single model across multiple scenes without retraining), our method also outperforms 4DLangSplat, gaining about 1% mIoU and 0.08% mAcc, showing strong cross-scene generalization with shared training weights.

**Time-Sensitive Language Queries.** Table 2 evaluates methods with time-sensitive queries, which require both spatial localization and temporal identification (e.g., "glass contains darker brown liquid" in Fig. 3). In the per-scene setting, 4DLangVGGT outperforms all baselines, exceeding 4DLangSplat by 0.03% Acc and 0.8% vIoU. Under the multi-video single-model setting, our model further improves temporal accuracy, surpassing per-scene models by 0.58% Acc and 1.68% vIoU. These results demonstrate that 4DLangVGGT more reliably captures object dynamics and semantic state changes, highlighting its strength in language–vision alignment and spatiotemporal consistency for dynamic 4D environments.

### 5.2.2 NEU3D DATASET

On the Neu3D dataset, which mainly consists of long-range videos where object dynamics are not prominent, we focus on time-agnostic language queries.

**Time-Agnostic Language Queries.** As shown in Table 3, our method (4DLangVGGT) achieves the best overall performance across all evaluated scenes, with an average of 87.41% mIoU and 99.41% mAcc, outperforming all baselines. Notably, compared to the second-best method 4DLangSplat, our approach yields consistent improvements in both mIoU and mAcc, demonstrating stronger spatial semantic grounding under this setting. In addition, we evaluate the more challenging multi-video

Table 3: Quantitative comparisons of **time-agnostic language queries** on the Neu3D dataset.

| Method | Per-scene | coffee martini | | cook spinach | | cut roasted beef | | Average | |
|---|---|---|---|---|---|---|---|---|---|
| | | mIoU(%) | mAcc(%) | mIoU(%) | mAcc(%) | mIoU(%) | mAcc(%) | mIoU(%) | mAcc(%) |
| Feature-3DGS (Zhou et al., 2024) | ✓ | 30.23 | 84.74 | 41.50 | 95.59 | 31.66 | 91.07 | 34.46 | 90.47 |
| Gaussian Grouping (Ye et al., 2024) | ✓ | 71.37 | 97.34 | 46.45 | 93.79 | 54.70 | 93.25 | 57.51 | 94.79 |
| LangSplat (Qin et al., 2024) | ✓ | 67.97 | 98.47 | 78.29 | 98.60 | 36.53 | 97.04 | 60.93 | 98.04 |
| 4DLangSplat (Li et al., 2025c) | ✓ | 85.16 | 99.23 | 85.09 | 99.38 | 85.32 | 99.28 | 85.19 | 99.30 |
| 4DLangVGGT (Ours) | ✓ | **87.59** | **99.40** | **86.93** | **99.52** | **87.72** | **99.32** | **87.41** | **99.41** |
| 4DLangVGGT (Ours) | ✗ | **85.51** | **99.35** | **85.25** | **99.47** | **86.17** | **99.30** | **85.64** | **99.37** |

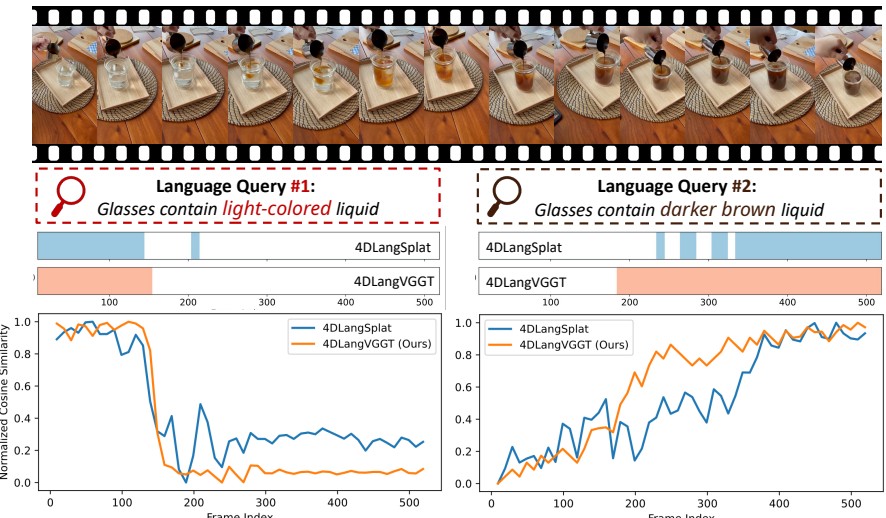

Figure 3: Qualitative results of time-sensitive language queries between 4DLangSplat and our 4DLangVGGT. Our 4DLangVGGT provides more accurate grounding compared to 4DLangSplat.

single-model setting, where a single model is jointly trained on multiple scenes and directly applied for inference without per-scene retraining. Even under this stricter condition, our model maintains strong performance, achieving 85.64% mIoU and 99.37% mAcc, which is close to the per-scene results. This highlights the efficiency and cross-scene generalization ability of our framework.

### 5.2.3 VISUALIZATION

To qualitatively assess the learned 4D semantic fields, we visualize both time-agnostic and time-sensitive query results in Fig. 3 and Fig. 4, respectively. For time-agnostic queries, our method produces sharper and more consistent masks than baseline methods, particularly in scenes with complex geometry or occlusions. For time-sensitive queries (as shown in Fig. 3), our framework can accurately capture critical semantic transitions, such as the moment when an object

Table 4: Ablation study of the RGB Head for reconstruction on Hypernerf dataset.

| RGB Head | Time-agnostic query | | Time-sensitive query | |
|---|---|---|---|---|
| | mIoU(%) | mAcc(%) | Acc(%) | vIoU(%) |
| ✓ | 83.99 | 98.67 | 91.44 | 74.74 |
| ✗ | 78.36 | 97.68 | 88.52 | 70.94 |

changes state or when an action begins (e.g., glasses contain darker brown liquid). In contrast, 4DLangSplat often struggles to detect such fine-grained changes, frequently producing temporally inconsistent masks or missing key state boundaries. These visualizations provide intuitive evidence that our method achieves superior semantic alignment with both spatial structures and temporal dynamics.

### 5.3 ABLATION STUDY

**Ablation Study on the RGB Head.** To investigate the contribution of the RGB reconstruction head in the Semantic Bridging Decoder (SBD), we conducted an ablation experiment in which

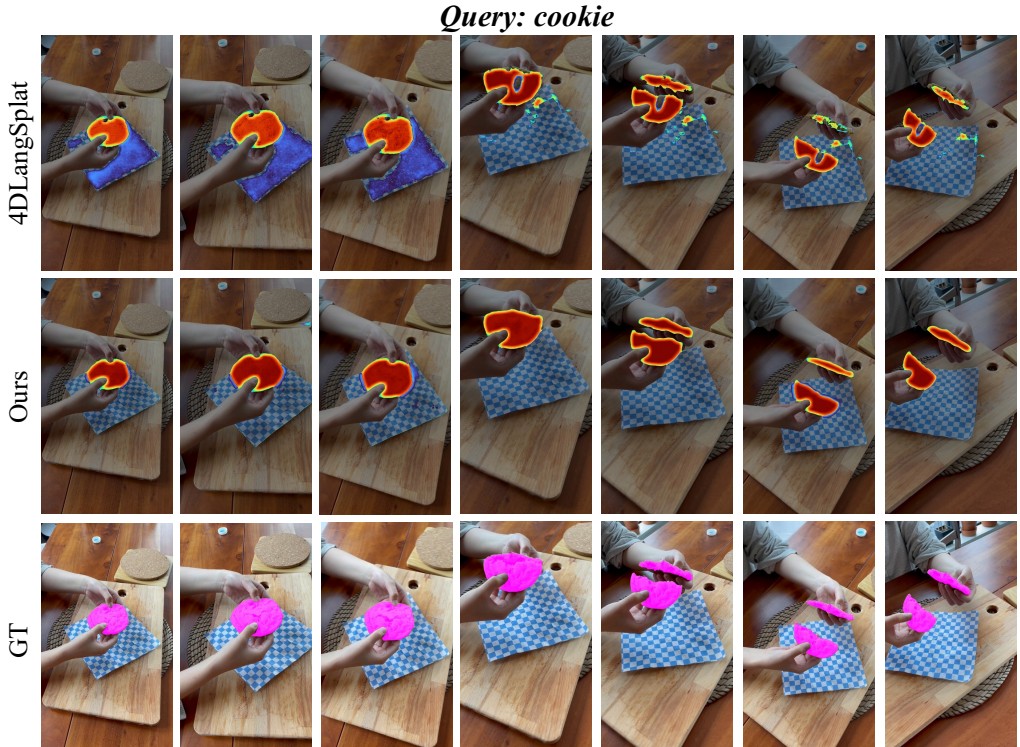

Figure 4: Comparison of time-agnostic query masks. The results demonstrate that our method consistently extracts accurate object masks in both intact and fragmented cookie scenarios, whereas 4DLangSplat exhibits degraded performance when handling fragmented cases.

the RGB Head was removed. The results, summarized in Table 4, demonstrate that removing the RGB Head leads to a noticeable drop (around 5% in IoU, 1-2% in Acc) in both time-agnostic and time-sensitive performance . This shows that the auxiliary reconstruction branch is essential for preserving appearance-level cues, which in turn strengthens semantic alignment and yields more accurate grounding.

**Ablation Study on the Architecture of Heads.**
We further compare different architectures for the Semantic and RGB Heads in Table 5. The UNet design achieves consistently better results than a simple MLP (improving by +0.95% mIoU, +1.16% mAcc, +2.06% Acc, and +2.15% vIoU). These gains highlight the benefit of UNet's hierarchical features for capturing fine-grained structures, leading to stronger spatial–temporal grounding than shallow alternatives.

Table 5: Ablation study of the different architectures for the RGB Head and Semantic Head.

| Architecture of Heads | Time-agnostic query | | Time-sensitive query | |
|---|---|---|---|---|
| | mIoU(%) | mAcc(%) | Acc(%) | vIoU(%) |
| UNet | 83.99 | 98.67 | 91.44 | 74.74 |
| MLP | 83.04 | 97.51 | 89.38 | 72.59 |

## 6 CONCLUSION

In this work, we introduced **4DLangVGGT**, a feed-forward framework that unifies geometry-aware 4D perception with language grounding for dynamic scene understanding. By leveraging the Semantic Bridging Decoder (SBD), the auxiliary RGB head and the joint supervision loss, our method effectively bridges low-level geometric cues and high-level semantic alignment, leading to more faithful and robust predictions. Extensive experiments on HyperNeRF and Neu3D demonstrate that 4DLangVGGT achieves strong performance and generalization without per-scene optimization, out-performing Gaussian-splatting baselines in both scalability and efficiency. These results highlight the potential of our framework as a step toward scalable, language-aware 4D semantic fields, paving the way for future extensions to larger-scale datasets and richer multimodal supervision.

**Ethics Statement.** We follow the ICLR Code of Ethics. Our work focuses on building 4D language fields for dynamic scene understanding and does not involve personal data, sensitive attributes, or identifiable human subjects beyond synthetic or publicly available datasets. Specifically, we conduct experiments on HyperNeRF and Neu3D, which are license-compliant research datasets containing synthetic or anonymized real-world scenes. We exclude any content that is violent, sexual, or otherwise harmful. The semantic annotations used in our work are automatically generated through established vision–language models (e.g., CLIP, SAM, DEVA) and do not involve manual collection of biometric or medical data. While our method learns semantic representations for 4D dynamic scenes, it does not involve identity-related, biometric, or personally sensitive data. The semantic features are derived from publicly available datasets and pretrained vision–language models, and do not enable identity reconstruction or manipulation. The release is restricted to research purposes only, with terms prohibiting harmful or deceptive uses. This study complies with all applicable policies on privacy, copyright, and research integrity.

**Reproducibility Statement.** We ensure reproducibility by providing a complete description of the 4DLangVGGT framework, including detailed formulations of the StreamVGGT-based geometry encoder, the Semantic Bridging Decoder, and the multi-objective training objective. Hyperparameters, loss weights, and dataset splits are clearly specified in the main text and appendix. We will release code, pretrained models to reproduce all reported results. Our supplementary materials include qualitative video visualizations on HyperNeRF and Neu3D. Together, these materials ensure that all results in the paper can be independently verified and extended.

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

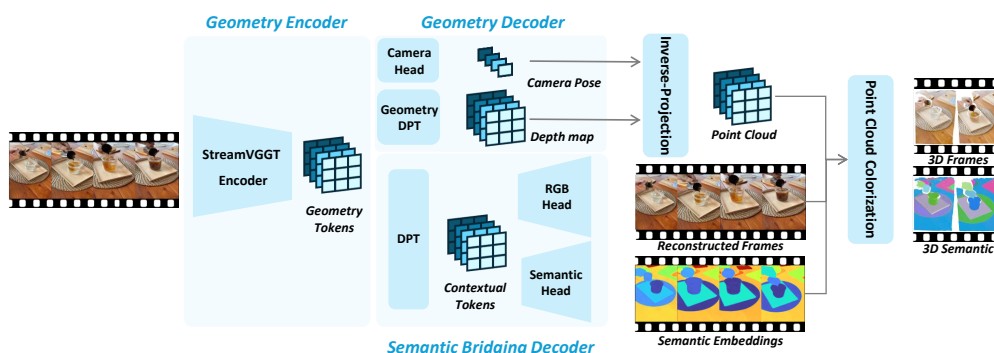

Figure 5: 4D inference pipeline of **4DLangVGGT**. Input video frames are processed by StreamVGGT to obtain geometry tokens. The Semantic Bridging Decoder predicts both RGB reconstructions and semantic embeddings, while the geometry decoder estimates depth maps and camera poses. Inverse-projection lifts them into a 3D point cloud, onto which the predicted RGB and semantics are colorized, yielding 3D frames and 3D semantic maps.

## A  MORE EXPERIMENTAL SETTINGS AND DETAILS

### A.1  IMPLEMENTATION DETAILS

**Input Resolution.**  StreamVGGT requires input resolutions to be multiples of 14. Therefore, we centrally crop all frames to the nearest multiple of 14, ensuring compatibility with the architecture while preserving the main scene content.

**Semantic Features.**  We extract feature embeddings from CLIP (512 dimensions) and E5 (4096 dimensions). To reduce dimensionality, two separate autoencoders are trained to compress these embeddings into 3-dimensions and 6-dimensions latent spaces, respectively.

**Training Hyperparameters.**  For the semantic loss $\mathcal{L}_{\mathrm{lang}}$ (Eq. 6), we set $\lambda_1 = 0.2$ and $\lambda_2 = 0.01$. For the reconstruction loss $\mathcal{L}_{\mathrm{rgb}}$ (Eq. 7), we set $\lambda_{\mathrm{img}} = 0.5$. We use the AdamW optimizer with an initial learning rate of $4 \times 10^{-5}$, weight decay of $1 \times 10^{-4}$, and gradient clipping at 1.0. A warm-up strategy of 20 epochs is applied, followed by either constant or cosine decay scheduling. The geometry encoder (StreamVGGT) is kept frozen, while the Semantic Bridging Decoder are trained.

**Metric.**  We adopt four standard metrics to evaluate both time-agnostic and time-sensitive querying. For the time-agnostic setting, **mean accuracy (mAcc)** measures the proportion of correctly predicted pixels, while **mean intersection-over-union (mIoU)** evaluates the overlap between predicted and ground-truth masks. For the time-sensitive setting, **accuracy (Acc)** reflects the ratio of correctly identified frames, and **video-level IoU (vIoU)** assesses spatial alignment within the predicted temporal segments. Together, these metrics provide a balanced evaluation of spatial precision and temporal consistency.

### A.2  INFERENCE IN 4D

At inference time, our framework takes a sequence of video frames as input and produces both geometry-aware reconstructions and semantic fields. The process is illustrated in Fig. 5.

First, the **StreamVGGT encoder** extracts spatio-temporal geometry tokens that capture the underlying 3D structure and temporal dynamics. These geometry tokens are fed into two parallel branches:

1. Our semantic bridging decoder predicts per-frame semantic embeddings aligned with natural language, while the RGB head reconstructs frames to ensure perceptual fidelity.

2. The depth head of StreamVGGT estimates dense depth maps, and the camera head of StreamVGGT predicts camera poses. Using these outputs, we perform inverse-projection to lift 2D pixels into a 3D point cloud.

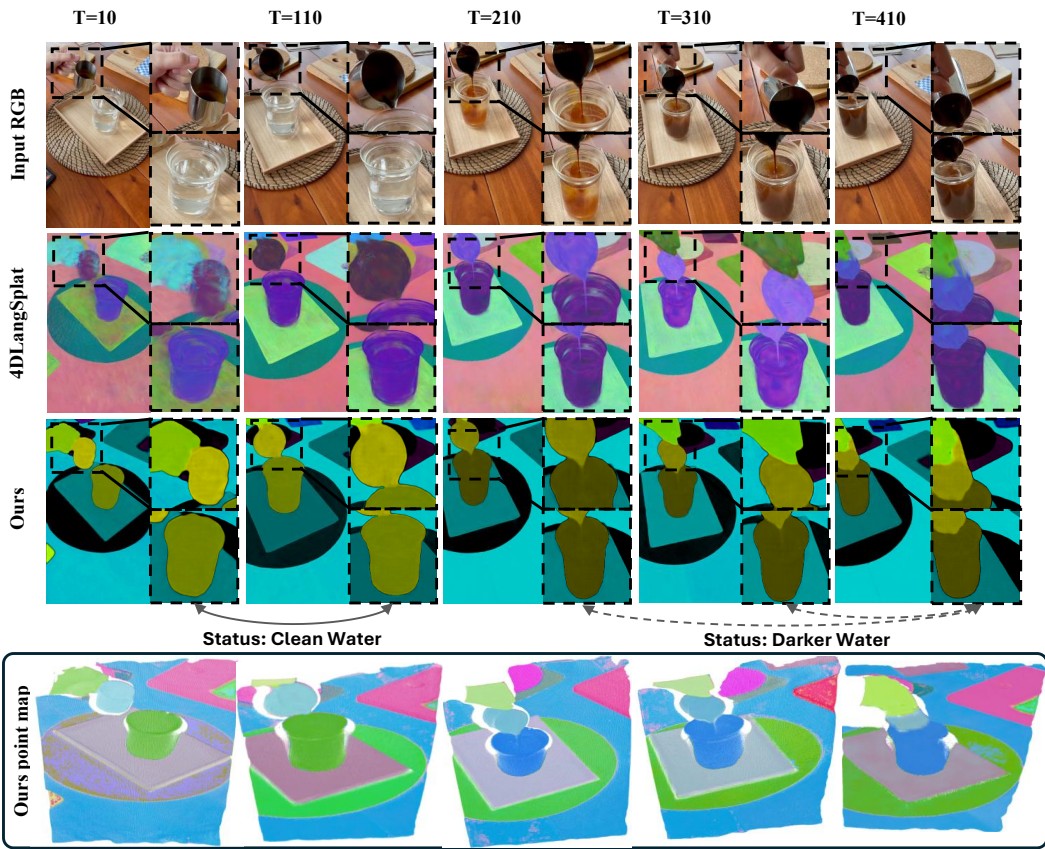

Figure 6: Additional qualitative comparison of language feature visualizations by our method and 4DLangSplat (Li et al., 2025c) like Fig. 1.

Finally, the predicted RGB values and semantic embeddings are colorized onto the 3D point map. This produces both **3D frames** (geometry with RGB appearance) and **3D semantic maps** (geometry with open-vocabulary semantics), enabling a unified 4D language field representation.

## B   MORE QUALITATIVE RESULTS

As shown in the visualizations in Fig. 6. The top part shows semantic visualizations learned by both methods, where our approach not only captures finer details (upper-right zoomed regions) but also demonstrates higher sensitivity to temporal state changes of objects (lower-right highlighted examples). The bottom part illustrates our method's learned semantic features projected onto 3D point clouds, providing a more interpretable view of spatiotemporal semantics in dynamic scenes.

## C   GENERALIZATION EXPERIMENT

### C.1   CROSS-DATASET GENERALIZATION EXPERIMENTS

As shown in the visualizations in Fig. 7, our method maintains stable reconstruction quality and produces coherent, artifact-free renderings on unseen datasets such as Objectron (Ahmadyan et al., 2021), even under substantial domain shifts. These results further demonstrate that our approach generalizes well beyond its training distribution.

### C.2   CROSS-QUERY GENERALIZATION EXPERIMENTS

We conducted a query-level generalization study, with results provided in Table 6. In this experiment, the original evaluation queries were replaced with semantically similar yet syntactically

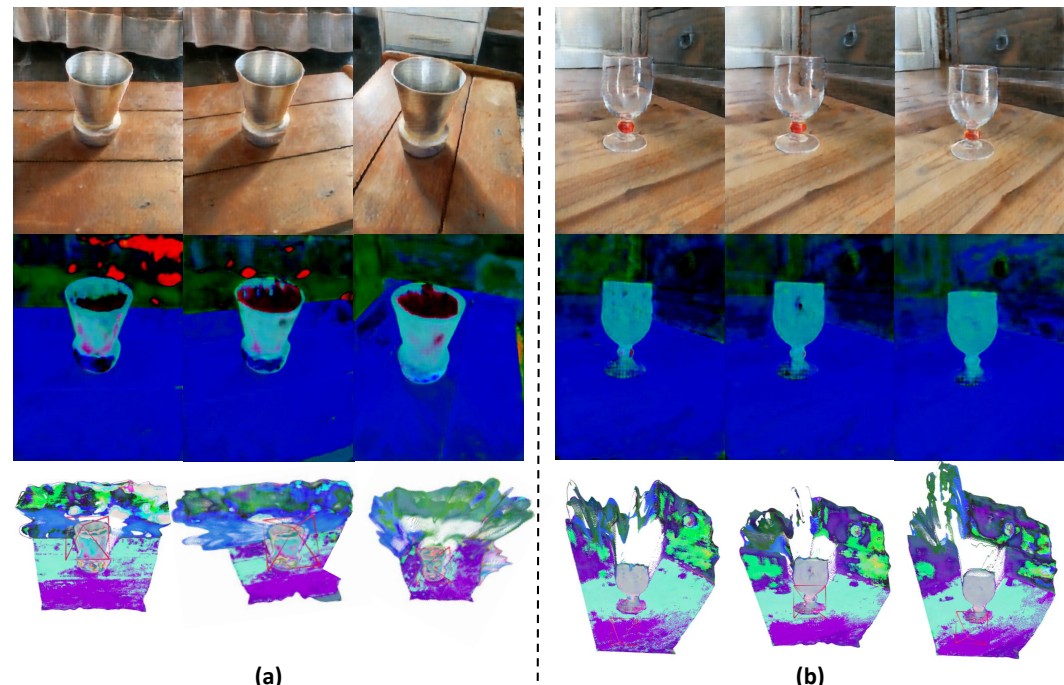

**(a)**       **(b)**

Figure 7: Additional robustness experiment. We train our method on the HyperNeRF dataset and evaluate it on videos from Objectron datasets to demonstrate cross-dataset generalization and visual robustness.

Table 6: Generalization experiments across different queries. We evaluated the performance of different time-sensitive queries on 4DLangSplat and 4DLangVGGT to explore their generalization capabilities across diverse queries.

| Query | americano | | | chick-chicken | | |
|---|---|---|---|---|---|---|
| | Raw query | Paraphrased query | $\Delta$ | Raw query | Paraphrased query | $\Delta$ |
| 4DLangSplat | 66.07 | 51.34 | -14.73 | 90.62 | 83.26 | -7.36 |
| 4DLangVGGT | 67.77 | 64.82 | -2.95 | 93.44 | 90.36 | -3.08 |

different expressions to test the model's robustness to linguistic variations. The results show that our model remains stable under such query changes and exhibits better cross-query generalization compared to 4DLangSplat. The modified queries used in this experiment are listed below:

- Query for americano
    - Raw query #1. Glasses contain light-colored liquid.
    - Raw query #2. Glasses contain dark brown liquid.
    - Paraphrased query #1. Glasses are filled with a light-colored liquid.
    - Paraphrased query #2. Glasses hold a deep brown-colored liquid.
- Query for chick-chicken
    - Raw query #1. Closed chicken container.
    - Raw query #2. Opened chicken container.
    - Paraphrased query #1. A chicken container that's sealed shut.
    - Paraphrased query #2. A container of chicken that is open.

Table 7: Ablation study of the DPT layer.

| DPT layer | Time-agnostic query | | Time-sensitive query | |
|---|---|---|---|---|
| | mIoU(%) | mAcc(%) | vIoU(%) | Acc(%) |
| ✗ | 80.36 | 96.49 | 72.15 | 89.37 |
| ✓ | 83.99 | 98.67 | 74.74 | 91.44 |
| Δ | + 3.63 | + 2.18 | + 2.59 | + 2.07 |

Table 8: Comparison of **time-agnostic language queries** under single-model training on HyperN-eRF. We report results from the paper non-joint training and joint training variant.

| Method | Joint | americano | | chick-chicken | | split-cookie | | torchocolate | | Average | |
|---|---|---|---|---|---|---|---|---|---|---|---|
| | | mIoU | mAcc | mIoU | mAcc | mIoU | mAcc | mIoU | mAcc | mIoU | mAcc |
| 4DLangVGGT (Ours) | ✗ | **82.46** | **98.36** | **86.91** | **98.88** | **91.44** | **98.87** | **75.15** | **98.57** | **83.99** | **98.67** |
| 4DLangVGGT (Ours, joint) | ✓ | 81.32 | 98.22 | 85.74 | 98.79 | 90.21 | 98.73 | 74.02 | 98.41 | 82.82 | 98.54 |

Table 9: Comparison of **time-sensitive language queries** under single-model training on HyperN-eRF. We report results from non-joint training and joint training variant.

| Method | Joint | americano | | chick-chicken | | split-cookie | | espresso | | Average | |
|---|---|---|---|---|---|---|---|---|---|---|---|
| | | Acc | vIoU | Acc | vIoU | Acc | vIoU | Acc | vIoU | Acc | vIoU |
| 4DLangVGGT (Ours) | ✗ | **90.03** | **67.77** | **97.81** | **93.44** | **95.76** | **84.02** | **82.86** | **52.06** | **91.44** | **74.74** |
| 4DLangVGGT (Ours, joint) | ✓ | 89.41 | 66.92 | 97.12 | 92.31 | 95.03 | 83.11 | 82.10 | 51.21 | 90.42 | 73.39 |

# D ADDITIONAL ABLATION STUDY

## D.1 DPT LAYER

As shown in Table 7, introducing the DPT layer yields clear improvements across both time-agnostic and time-sensitive evaluations: **+3.63%** mIoU and **+2.18%** mAcc for time-agnostic queries, and **+2.59%** vIoU and **+2.07%** Acc for time-sensitive queries. These results demonstrate that DPT's contextual modeling significantly enhances semantic discrimination while improving both spatial and temporal alignment.

## D.2 JOINTLY TRAINING IN TIME-AGNOSTIC AND TIME SENSITIVE

As shown in Table 8 and Table 9, joint training leads to a consistent but mild drop 1.3% in time-agnostic and 1.1% in time-sensitive compared with the non-joint train in the paper. This confirms that the two types of semantic supervision indeed encode different objectives: static object semantics vs. dynamic state transitions, and forcing them into a shared representation introduces functional interference. These new results further justify our architectural design choice of using decoupled branches, each specialized for its intended type of semantic signal.

# E LIMITATION AND FUTURE WORKS

While our work introduces **4DLangVGGT**, the first unified Transformer-based framework for 4D language fields, there are still several limitations to address. First, our experimental validation is limited to HyperNeRF and Neu3D, which contain only a small number of dynamic scenes. Although these benchmarks are standard in prior literature, they do not fully reflect the scale and diversity of real-world environments. Consequently, the generalization of our framework to more complex and large-scale settings remains to be thoroughly explored.

In future work, we plan to scale up our approach to substantially larger and more diverse datasets, aiming to rigorously evaluate both efficiency and robustness under real-world conditions. We will explore improving the fine-grained and precision of dynamic semantic supervision, which is inspired by recent Mask Grounding approaches (Chng et al., 2024), which demonstrate the effectiveness

of achieving fine-grained alignment between linguistic expressions and localized visual regions. Furthermore, we envision the development of a domain-specific large model for 4D language fields, which can serve as a foundation model for embodied AI, AR/VR, and open-vocabulary dynamic scene understanding. Such a model would unify semantic reasoning and geometric perception at scale, potentially enabling new applications that go beyond current scene-level benchmarks.

