# OpenReview forum: "4DLangVGGT: 4D Language Visual Geometry Grounded Transformer"
_ICLR.cc/2026/Conference — Submitted to ICLR 2026_

### Official Review · Reviewer_1e88 · 2025-10-21

**Soundness:** 3
**Presentation:** 3
**Contribution:** 2
**Rating:** 4
**Confidence:** 4

**Summary:**

This paper proposes 4DLangVGGT, a Transformer-based feed-forward framework for 4D language grounding in dynamic scenes. The method addresses the limitations of existing approaches, which typically rely on per-scene optimization (e.g., Gaussian Splatting) and thus suffer from poor scalability and generalization. 4DLangVGGT uses a frozen, pre-trained StreamVGGT as a geometry encoder and introduces a novel, trainable Semantic Bridging Decoder (SBD). The SBD maps spatio-temporal geometric features into a language-aligned semantic space. A key advantage of this architecture is its ability to be trained jointly across multiple scenes and applied directly at inference without per-scene optimization. Experiments on the HyperNeRF and Neu3D datasets show that 4DLangVGGT achieves state-of-the-art performance, outperforming previous methods in both per-scene and multi-scene training setups.

**Strengths:**

1. **Interesting Problem**: The paper tackles a critical and practical limitation of current 4D semantic field models: their reliance on costly and non-generalizable per-scene optimization. The proposed feed-forward, generalizable framework is a valuable contribution toward scalable 4D scene understanding.

2. **Strong Performance**: The method achieves state-of-the-art results on two standard benchmarks (HyperNeRF and Neu3D). It consistently outperforms the primary baseline, 4DLangSplat, in both time-agnostic and time-sensitive query settings.

3. **Well-motivated Architecture**: The core idea of decoupling the architecture into a frozen geometry encoder (StreamVGGT) and a trainable Semantic Bridging Decoder (SBD) is sound. This design efficiently leverages powerful pre-trained geometry representations while focusing training on the novel task of aligning geometry with language semantics.

**Weaknesses:**

1. **Missing Efficiency Analysis**: The central motivation for the feed-forward design is to avoid "costly per-scene optimization" and achieve "efficiency". However, the paper provides no quantitative comparison of efficiency. There is no analysis of training time, inference speed (e.g., FPS), or memory footprint compared to the 4DLangSplat baseline. This is a critical gap; without this data, the claimed efficiency advantage is unsubstantiated.

2. **Limited Scale of Generalization**: The core claim of generalization is tested on very small datasets. The paper notes joint training across 6 scenes in HyperNeRF and 6 scenes in Neu3D , and the authors acknowledge this limitation in the appendix. Demonstrating generalization on such a small number of scenes is not a robust validation of a "scalable" framework intended for "large-scale deployment".

3. **Missing Ablation on Encoder**: The choice to keep the StreamVGGT encoder frozen is a major architectural decision that is not ablated. Freezing is justified for efficiency, but it may also create a bottleneck by preventing the geometry features from adapting to the semantic task. An ablation comparing a frozen encoder versus a fine-tuned one is necessary to understand this trade-off and validate the design choice.

4. **Missing Ablation on Semantic Loss**: The semantic supervision is complex, combining time-agnostic CLIP embeddings with time-sensitive MLLM-generated embeddings. The paper fails to ablate the contributions of these two components. It is unclear how much performance is gained from the expensive MLLM-based dynamic supervision versus using only the simpler static CLIP features. This ablation is essential to justify the complexity of the ground-truth generation pipeline.

5. **Marginal Per-Scene Gains**: While the method is SOTA, its performance gains in the direct per-scene comparison (which is the baseline's intended setting) are sometimes marginal. For instance, on HyperNeRF for time-sensitive queries, the improvement over 4DLangSplat is only +0.03% in Accuracy and +0.8% in vIoU. This suggests that while its generalization is novel, its raw performance in a constrained setting is not a significant leap.

6. **Omission of Related Work**: *Mask Grounding for Referring Image Segmentation* (Chng et al., CVPR 2024) introduces a mask-grounding task for RIS—predicting masked textual tokens given visual features and a segmentation mask—which directly overlaps with your time-agnostic/time-sensitive mask-conditioned language supervision (i.e., region-aligned per-pixel semantic targets inside SAM/DEVA masks); this prior work should be discussed and, where feasible, compared against to clearly situate novelty and empirical gains.

**Questions:**

See weaknesses.

---

> ### Author Response · Authors · 2025-11-29
> **Rebuttal by Authors (Part 1/3)**
>
> We sincerely thank the reviewer for the constructive comments and suggestions. We hope that our responses below adequately address the concerns raised.
>
> `Question 1: Missing Efficiency Analysis.`
>
> ##### *Table 1: GPU time comparisons for training. Total\* means 4DLangSplat sums all single-scene training times, while 4DLangVGGT is the cross-scene training time.*
>
> | Scene         | **4DLangSplat**              |                      | **4DLangVGGT (Ours)**        |                      |
> |---------------|------------------------------|----------------------|------------------------------|----------------------|
> |               | Time-agnostic (hour)         | Time-sensitive (hour)| Time-agnostic (hour)         | Time-sensitive (hour)|
> | Americano     | 1.10                         | 0.35                 | -                            | -                    |
> | Chichchicken  | 1.05                         | 0.87                 | -                            | -                    |
> | Espresso      | 1.25                         | 0.86                 | -                            | -                    |
> | Split-cookie  | 0.99                         | 0.35                 | -                            | -                    |
> | Torchocolate  | 1.02                         | 0.35                 | -                            | -                    |
> | Keyboard      | 1.07                         | 0.85                 | -                            | -                    |
> | **Total\***   | **6.48**                     | **3.63**             | **6.15**                     | **3.03**             |
>
> As shown in Table 1, when comparing the training time across 6 scenes, 4DLangSplat requires per-scene optimization (taking 6.48 hours/3.63 hours), while 4DLangVGGT, by constructing a cross-scene generalizable model, needs only a single training session (totaling 6.15 hours/3.03 hours) to cover all scenes. Although the total training times are currently comparable at this dataset scale, 4DLangVGGT already demonstrates advantages in amortized training cost. Regarding inference efficiency, Table 2 shows that 4DLangSplat achieves 31.1 FPS on the Americano scene based on its pre-optimized Gaussian splatting representation, whereas 4DLangVGGT, with its feed-forward backbone requiring per-frame computation, currently reaches 2.85 FPS.
>
> ##### *Table 2: Inference speed comparisons of time-agnostic querying (FPS)*
>
> | Scene         | **4DLangSplat**        |                 | **4DLangVGGT**         |                 |
> |---------------|-------------------------|-----------------|-------------------------|-----------------|
> |               | Time-agnostic           | Time-sensitive  | Time-agnostic           | Time-sensitive  |
> | Americano     | 31.1                    | 5.87            | 2.85                    | 1.9             |
> | Chichchicken  | 30.2                    | 5.69            | 2.68                    | 1.4             |
>
> We would like to emphasize that our core efficiency contribution lies in completely eliminating the need for per-scene optimization. While the current inference speed is slower, as the number of application scenarios increases, the "train once, apply everywhere" characteristic of 4DLangVGGT will demonstrate significant scalability advantages—its total training cost remains constant, while the cost of per-scene methods grows linearly with the number of scenes. We are actively working on optimizing the network architecture to further improve inference efficiency.
>
> `Question 2: Limited Scale of Generalization.`
>
> Thank you for this comment. We agree that the scale of the training and evaluation datasets (HyperNeRF and Neu3D) is indeed limited, and we have explicitly acknowledged this point in the Limitations section. That said, our primary contribution lies in **proposing a transformer-based feed-forward unified framework for 4D language grounding**, which is fundamentally different from previous per-scene optimization approaches such as 4D LangSplat. Even on these smaller-scale benchmarks, our method has successfully demonstrated the feasibility and advantages of a generalizable architecture, achieving competitive performance without scene-specific retraining.
>
> In future work, we plan to scale up the training data to further improve the model's generalization ability across more diverse and complex 4D scenes. We appreciate the reviewer's insightful feedback, which will help guide our next steps.

---

> ### Author Response · Authors · 2025-11-29
> **Rebuttal by Authors (Part 2/3)**
>
> `Question 3: Missing Ablation on Encoder.`
>
> ##### *Table 3: Quantitative comparisons of the frozen and fine-tuned StreamVGGT*
>
> | Status                 | **americano**        |               | **chick-chicken**     |               | **split-cookie**      |               | **Average**           |               |
> |------------------------|----------------------|---------------|------------------------|---------------|------------------------|---------------|------------------------|---------------|
> |                        | mIoU(%)              | mAcc(%)       | mIoU(%)                | mAcc(%)       | mIoU(%)                | mAcc(%)       | mIoU(%)                | mAcc(%)       |
> | Frozen StreamVGGT      | 82.46                | 98.36         | 86.91                  | 98.88         | 91.44                  | 98.87         | 83.99                  | 98.67         |
> | Fine-tuned StreamVGGT  | 84.67                | 98.45         | 87.30                  | 98.85         | 91.82                  | 98.96         | 84.66                  | 98.74         |
> | Δ                      | +2.21                | +0.09         | +0.39                  | -0.03         | +0.38                  | +0.09         | +0.67                  | +0.07         |
>
> We thank the reviewer for for raising this point. As shown in Table 3, fine-tuning StreamVGGT yields consistent but relatively small performance gains: the mIoU increases by 2.21, 0.39, and 0.38 points across the three scenes, resulting in a 0.67 point improvement on average, while mAcc varies within a narrow ±0.1 range. Although these results confirm that fine-tuning is beneficial, the magnitude of improvement remains limited. This is primarily due to the small scale of our training set,  extensive fine-tuning under such data scarcity can easily lead to overfitting and diminish the geometric priors already encoded in the pretrained backbone. Consequently, the model’s performance improves, but the extent of enhancement is naturally constrained by the available data.
>
> `Question 4: Missing Ablation on Semantic Loss`
>
> We thank the reviewer for this valuable feedback. We fully agree that the individual contributions of the two semantic supervision signals were not sufficiently analyzed in the paper.
>
> In our design, the two types of supervision serve distinct objectives: the time-agnostic semantic supervision is designed to learn static semantic representations of objects in the scene, while the time-sensitive semantic supervision focuses on modeling the dynamic semantic evolution of objects. Accordingly, we implemented two separate decoder branches, parameter-independent and functionally decoupled, each dedicated to a specific type of query task. For instance, on the HyperNeRF dataset, time-agnostic language queries in Table 1 of the main paper exclusively use the time-agnostic branch, whereas time-sensitive queries in Table 2 of the main paper activate only the time-sensitive branch.
>
> ##### *Table 4: Comparison of time-agnostic language queries under single-model training on HyperNeRF.*
>
> | Method                | Joint | americano mIoU | americano mAcc | chick mIoU | chick mAcc | cookie mIoU | cookie mAcc | torchocolate mIoU | torchocolate mAcc | Avg mIoU | Avg mAcc |
> |----------------------|:-----:|:--------------:|:---------------:|:----------:|:-----------:|:-----------:|:------------:|:------------------:|:-------------------:|:---------:|:---------:|
> | 4DLangVGGT (Ours)    | ×     | **82.46** | **98.36** | **86.91** | **98.88** | **91.44** | **98.87** | **75.15** | **98.57** | **83.99** | **98.67** |
> | 4DLangVGGT (joint)   | ✓     | 81.32 | 98.22 | 85.74 | 98.79 | 90.21 | 98.73 | 74.02 | 98.41 | 82.82 | 98.54 |
>
>
> ##### *Table 5: Comparison of time-sensitive language queries under single-model training on HyperNeRF.*
>
> | Method                | Joint | americano Acc | americano vIoU | chick Acc | chick vIoU | cookie Acc | cookie vIoU | espresso Acc | espresso vIoU | Avg Acc | Avg vIoU |
> |----------------------|:-----:|:--------------:|:---------------:|:----------:|:-----------:|:-----------:|:------------:|:-------------:|:---------------:|:---------:|:----------:|
> | 4DLangVGGT (Ours)    | ×     | **90.03** | **67.77** | **97.81** | **93.44** | **95.76** | **84.02** | **82.86** | **52.06** | **91.44** | **74.74** |
> | 4DLangVGGT (joint)   | ✓     | 89.41 | 66.92 | 97.12 | 92.31 | 95.03 | 83.11 | 82.10 | 51.21 | 90.42 | 73.39 |
>
>
> Inspired by the reviewer’s comment, we are now conducting further experiments to compare the performance of jointly training both branches versus training them separately. As shown in Table 4 and Table 5, this indicates that our pipeline possesses strong inherent robustness, and its performance can be further improved with larger training datasets

---

> ### Author Response · Authors · 2025-11-29
> **Rebuttal by Authors (Part 3/3)**
>
> `Question 5: Marginal Per-Scene Gains`
>
> The limited improvements actually reflects a fundamental trade-off in our methodological design: balancing universal scene adaptability against optimal single-scene performance. It should be noted that 4DLangSplat, as a method specifically optimized for individual scenes, can indeed achieve near-saturation performance in particular scenes. In contrast, the core objective of our work is to achieve cross-scene generalization capability. By constructing the unified feed-forward model 4DLangVGGT, we have for the first time realized multi-scene 4D open-vocabulary querying without requiring any per-scene optimization. More importantly, within this more challenging multi-scene setting, our method still achieves performance comparable to or even better than specifically optimized per-scene methods. This convincingly demonstrates that our geometry-centric design not only maintains competitive baseline performance but, more crucially, enables effective cross-scene generalization, thereby laying the foundation for subsequent large-scale applications.
>
> We believe this technical approach, achieving generalization while maintaining performance competitiveness, will provide a more valuable solution for the practical deployment of 4D scene understanding.
>
> `Question 6: Omission of Related Work`
>
> We will add a discussion of Mask Grounding. While there are superficial similarities in using mask-based semantic supervision, there are fundamental differences in task formulation and technical scope. Mask Grounding focuses on 2D referring image segmentation in static images, aiming to predict masked textual tokens from visual features and segmentation masks. In contrast, our work targets 4D open-vocabulary querying in dynamic scenes, requiring reasoning about geometry, temporal dynamics, and camera motion. Specifically, our time-agnostic supervision builds cross-scene universal semantic representations, while time-sensitive supervision models semantic evolution in dynamic environments. Inspired by recent Mask Grounding approaches [1], which demonstrate the effectiveness of achieving fine-grained alignment between linguistic expressions and localized visual regions. We will explore improving the fine-grained and precision of dynamic semantic supervision in the future.
>
> In 3D/4D language grounding, the LangSplat series represents the main prior work. We have comprehensively collected relevant literature and compared against the strongest available baselines. Our key innovation lies in integrating geometric priors with dynamic semantic modeling to provide a unified solution for open-world 4D scene understanding.
>
> [1] Chng Y X, Zheng H, Han Y, Gao H, Mask grounding for referring image segmentation[C]//Proceedings of the IEEE/CVF Conference on Computer Vision and Pattern Recognition (CVPR), 2024, pp. 26573-26583

---

### Official Review · Reviewer_q5cc · 2025-10-29

**Soundness:** 2
**Presentation:** 2
**Contribution:** 2
**Rating:** 4
**Confidence:** 3

**Summary:**

This paper introduces 4DLangVGGT, a Transformer-based, feed-forward unified framework for 4D language grounding. The core idea is to leverage the strong geometric priors of StreamVGGT and augment it with two heads to enhance semantic modeling. The Semantic Head is supervised by frozen foundation models (SAM, CLIP, and an LLM) to strengthen object-level semantics and temporal consistency. The RGB Head reconstructs per-frame images to preserve perceptual fidelity. Experiments show improvements on both time-agnostic and time-sensitive benchmarks.

**Strengths:**

1. The framework is clearly structured: semantic capacity is enhanced by adding lightweight heads to a VGGT backbone with targeted semantic supervision.
2. The paper is well written with clear figures, facilitating readability.
3. Extensive experiments are conducted under both time-agnostic and time-sensitive settings, with reasonable ablations and visualizations.

**Weaknesses:**

1. The main contribution, semantic supervision in the Semantic Head, is under-analyzed. The paper lacks a thorough breakdown of how the two supervision signals contribute individually, as well as component-wise ablations within the time-sensitive semantic supervision.
2. Compared to 4DLangSplat, the performance gains appear modest: Tables 1 and 3 seem nearly saturated with +0.1–0.2 improvements, and Table 2 reports only +1.7. (I may be less familiar with these benchmarks; please correct me if I’ve misread.)
3. Using LLM-based caption features for time-sensitive supervision may be questionable: captions provide global semantics without sufficiently fine-grained temporal signals, and cross-modal alignment could be nontrivial.

**Questions:**

1. Please provide ablations isolating the two semantic supervision signals, and component-wise ablations for the time-sensitive semantic supervision.
2. Please explain why the improvements over 4DLangSplat are limited, despite the added supervisory signals.
3. In light of Weakness #3, please justify the use of LLM caption features for time-sensitive supervision or provide experiments demonstrating their effectiveness.
4. The DPT component lacks empirical analysis, please include experiments quantifying its contribution.
5. Can you report inference speed comparisons?

---

> ### Author Response · Authors · 2025-11-29
> **Rebuttal By Authors (Part 1/3)**
>
> Thank you for the thoughtful review. We have carefully addressed each comment and updated the manuscript to improve clarity and quality.
>
> `Question 1: Ablation study for two supervision signals in the semantic head.`
>
> We thank the reviewer for this valuable feedback. We fully agree that the individual contributions of the two semantic supervision signals were not sufficiently analyzed in the paper.
>
> In our design, the two types of supervision serve distinct objectives: the time-agnostic semantic supervision is designed to learn static semantic representations of objects in the scene, while the time-sensitive semantic supervision focuses on modeling the dynamic semantic evolution of objects. Accordingly, we implemented two separate decoder branches, parameter-independent and functionally decoupled, each dedicated to a specific type of query task. For instance, on the HyperNeRF dataset, time-agnostic language queries in Table 1 of the main paper exclusively use the time-agnostic branch, whereas time-sensitive queries in Table 2 of the main paper activate only the time-sensitive branch.
>
> ##### *Table 1: Comparison of time-agnostic language queries under single-model training on HyperNeRF.*
>
> | Method                | Joint | americano mIoU | americano mAcc | chick mIoU | chick mAcc | cookie mIoU | cookie mAcc | torchocolate mIoU | torchocolate mAcc | Avg mIoU | Avg mAcc |
> |----------------------|:-----:|:--------------:|:---------------:|:----------:|:-----------:|:-----------:|:------------:|:------------------:|:-------------------:|:---------:|:---------:|
> | 4DLangVGGT (Ours)    | ×     | **82.46** | **98.36** | **86.91** | **98.88** | **91.44** | **98.87** | **75.15** | **98.57** | **83.99** | **98.67** |
> | 4DLangVGGT (joint)   | ✓     | 81.32 | 98.22 | 85.74 | 98.79 | 90.21 | 98.73 | 74.02 | 98.41 | 82.82 | 98.54 |
>
>
> ##### *Table 2. Comparison of time-sensitive language queries under single-model training on HyperNeRF.*
>
> | Method                | Joint | americano Acc | americano vIoU | chick Acc | chick vIoU | cookie Acc | cookie vIoU | espresso Acc | espresso vIoU | Avg Acc | Avg vIoU |
> |----------------------|:-----:|:--------------:|:---------------:|:----------:|:-----------:|:-----------:|:------------:|:-------------:|:---------------:|:---------:|:----------:|
> | 4DLangVGGT (Ours)    | ×     | **90.03** | **67.77** | **97.81** | **93.44** | **95.76** | **84.02** | **82.86** | **52.06** | **91.44** | **74.74** |
> | 4DLangVGGT (joint)   | ✓     | 89.41 | 66.92 | 97.12 | 92.31 | 95.03 | 83.11 | 82.10 | 51.21 | 90.42 | 73.39 |
>
>
> Inspired by the reviewer’s comment, we are now conducting further experiments to compare the performance of jointly training both branches versus training them separately. As shown in Table 1 and Table 2, this indicates that our pipeline possesses strong inherent robustness, and its performance can be further improved with larger training datasets
>
> `Question 2: Explaining modest improvements over 4DLangSplat.`
>
> Thank you for this thoughtful observation. You are correct that the performance gains over 4DLangSplat in certain metrics appear modest. This is, in fact, consistent with our expectations and highlights a key difference in the objectives of the two methods.
>
> As a per-scene optimized model, 4DLangSplat is inherently designed to achieve near-saturation performance on each specific scene. In contrast, our 4DLangVGGT is a general-purpose, feed-forward model trained across multiple scenes, intended for scalable cross-scene generalization without test-time optimization. The fact that our approach, under this more challenging and practical setting, still matches or slightly surpasses the per-scene upper bound performance of 4DLangSplat demonstrates its effectiveness. The marginal but consistent improvements reflect not only the strength of our unified framework but, more importantly, its potential for open-world scalability beyond the training set.

---

> ### Author Response · Authors · 2025-11-29
> **Rebuttal By Authors (Part 2/3)**
>
> `Question 3: Justifying LLM-Based caption features for time-sensitive supervision.`
>
> We thank the reviewer for their insightful comments. You rightly identified two key challenges in using LLM-generated descriptions for time-sensitive supervision: the lack of sufficiently fine-grained temporal signals and the potential difficulty of cross-modal alignment.
>
> ##### *Table 3. Generalization experiments across different queries. We evaluated the performance of different time-sensitive queries on 4DLangSplat and 4DLangVGGT to explore their generalization capabilities across diverse queries.*
>
> | Query          |            |  **americano** |       |          |   **chick-chicken**   |      |
> |----------------|-----------|-------------------|--------|-----------|-------------------|--------|
> |                | Raw Query | Paraphrased Query | Δ      |  Raw Query | Paraphrased Query | Δ     |
> | 4DLangSplat    | 66.07     | 51.34             |-14.73 | 90.62     | 83.26             |-7.36  |
> | 4DLangVGGT     | 67.77     | 64.82             |-2.95  | 93.44     | 90.36             |-3.08  |
>
> In the experiments conducted by 4DLangSplat (Table 3 and Section 4.2), despite these theoretical limitations, the MLLM+LLM description-based approach still significantly outperformed CLIP-based methods in time-sensitive query tasks on the HyperNeRF dataset. This suggests that object-level temporal semantic consistency, even if not finely-grained, provides critical information for improving 4D language grounding performance. We believe that the high-level state transition semantics in the descriptions (e.g., `picking up a cup` to `putting down the cup`) compensate for the lack of frame-level granularity.
> Meanwhile, 4DLangSplat alleviated the cross-modal alignment challenge to some extent by projecting LLM embeddings and visual features into a unified semantic space. Our work adopts this validated supervision signal and integrates it into a feed-forward geometric backbone to explore its potential in a generalizable architecture. We fully acknowledge the importance of fine-grained temporal supervision and will consider it an important direction for future work.
>
> `Question 4: Ablation study for DPT layer.`
>
> ##### *Table 4. Ablation study of the DPT layer.*
>
> | DPT Layer | **Time-agnostic query** |          | **Time-sensitive query** |          |
> |-----------|--------------------------|----------|---------------------------|----------|
> |           | mIoU(%)                  | mAcc(%)  | vIoU(%)                   | Acc(%)   |
> | ×         | 80.36                    | 96.49    | 72.15                     | 89.37    |
> | ✓         | 83.99                    | 98.67    | 74.74                     | 91.44    |
> | Δ         | +3.63                    | +2.18    | +2.59                     | +2.07    |
>
> Thank you for the helpful suggestion. We have added an ablation experiment that evaluates the effect of the DPT layer within the Semantic Bridging Decoder.
>
> As shown in Table 4, introducing the DPT layer yields clear improvements across both time-agnostic and time-sensitive evaluations:
> **+3.63%** mIoU and **+2.18%** mAcc for time-agnostic queries, and
> **+2.59%** vIoU and **+2.07%** Acc for time-sensitive queries.
> These results demonstrate that DPT’s contextual modeling significantly enhances semantic discrimination while improving both spatial and temporal alignment.

---

> ### Author Response · Authors · 2025-12-03
> **Rebuttal By Authors (Part 3/3)**
>
> `Question 5: Inference speed comparisons.`
>
> ##### *Table 5. Inference speed comparisons of time-agnostic querying (FPS)*
>
> | Scene         | **4DLangSplat**        |                 | **4DLangVGGT**         |                 |
> |---------------|-------------------------|-----------------|-------------------------|-----------------|
> |               | Time-agnostic           | Time-sensitive  | Time-agnostic           | Time-sensitive  |
> | Americano     | 31.1                    | 5.87            | 2.85                    | 1.9             |
> | Chichchicken  | 30.2                    | 5.69            | 2.68                    | 1.4             |
>
> We have added inference speed comparisons between 4DLangVGGT and 4DLangSplat on the HyperNeRF dataset in Table 5. As shown in the results, 4DLangSplat achieves higher frames per second (FPS) during inference, which primarily benefits from its pre-optimized, scene-specific 4D Gaussian splatting representation that enables highly efficient rendering. In comparison, 4DLangVGGT employs a Transformer-based feed-forward backbone and reconstructs the semantic field frame by frame, resulting in relatively slower inference speed.
>
> However, we would like to emphasize that the core advantage of 4DLangVGGT lies in its cross-scene generalization capability. As a generalizable architecture, our model only requires a single training session and can then be directly applied to new scenes without any scene-specific optimization. In contrast, 4DLangSplat, as a per-scene optimization method, requires separate training of both the 4D Gaussian field and the language field for each new scene. Its overall efficiency decreases significantly as the number of scenes increases. In the revised appendix, we added generalization experiments on the untrained dataset Objectron[1], with visualizations presented in Figure 7. Meanwhile, 4DLangSplat cannot directly reason in untrained scenarios.
>
> We believe that as the scale of training data expands, the generalizable design of 4DLangVGGT will demonstrate stronger scalability and practical value. While the current focus of our work is to validate the feasibility of the method in cross-scene settings, we plan to further optimize inference efficiency in future work and verify its scaling potential on larger-scale datasets.
>
> [1] Ahmadyan A, Zhang L, Ablavatski A, et al. Objectron: A large scale dataset of object-centric videos in the wild with pose annotations[C]//Proceedings of the IEEE/CVF conference on computer vision and pattern recognition. 2021: 7822-7831.

---

### Official Review · Reviewer_bJWm · 2025-10-31

**Soundness:** 3
**Presentation:** 3
**Contribution:** 3
**Rating:** 8
**Confidence:** 4

**Summary:**

The goal of this paper is to introduce a transformer-based approach for constructing 4D semantic fields.

The proposed approach aligns the spatio-temporal representation of dynamic scenes with semantic fields through time-sensitive and time-agnostic semantic supervision. The pipeline consists of a 4D visual geometry encoder (frozen StreamVGGT) and a DPT-based semantics alignment decoder with RGB and semantic heads. It is trained by minimizing joint reconstruction and the semantic objective.

The contributions of this work are as follows:
1) state-of-the-art approach for open-vocabulary 4D understanding;
2) multi-objective training pipeline;
3) ablation study of the architecture.

**Strengths:**

1) State-of-the-art results on open-vocabulary 4D scene understanding benchmarks.
2) Aproach does not require per-scene optimization.
3) Well-structured, easy to read, and follow manuscript.

**Weaknesses:**

1) Description of how open-vocabulary 4D querying that is missing in the manuscript.
2) As mentioned in limitations section training and evaluation is conducted on small dataset (HyperNeRF and Neu3D).
3) Moreover training and evaluation is performed on the same data. In other words the approach was evaluated on the data it was trained on. Thus though approach supports multy-scene training the cross scene/cross dataset generalization is not performed.

**Questions:**

1. How how open-vocabulary 4D querying is done?
2. L503-504: Supplementary materials are not attached.
3. L058: "achieving both efficiency and strong generalization" Strong generalization to training data (across training scenes)? Have you performed cross scene/cross dataset evaluation (training on the one sсene/dataset and evaluating on the other)?

---

> ### Author Response · Authors · 2025-11-29
> **Rebuttal By Authors (Part 1/2)**
>
> We sincerely thank the reviewer for the constructive comments and suggestions. We hope that our responses below adequately address the concerns raised.
>
> `Question 1: How open-vocabulary 4D querying`
>
> Our 4D open-vocabulary querying pipeline follows the core principle of `2D semantic matching, 3D geometric transformation, 4D Spatio-Temporal Integration' and is implemented in three main steps:
>
> 1. **2D Semantic Matching**: First, StreamVGGT extracts geometric features from video frames, which are then processed by the Semantic Bridging Decoder to generate pixel-level semantic features. Simultaneously, text prompts are encoded using CLIP or MLLM. By computing the similarity between the semantic features and the text embeddings, 2D language query results are obtained for each frame.
>
> 2. **3D Geometric Reconstruction**: In parallel, the geometric decoder of StreamVGGT predicts the depth map and camera pose for each frame, providing the geometric foundation for the 2D-to-3D conversion.
>
> 3. **4D Spatio-Temporal Integration**: The 2D language query results are combined with the geometric information, and matched pixel locations are transformed into 3D query point clouds via inverse projection. This finally constructs a temporally consistent sequence of 3D query frames, achieving open-vocabulary 4D scene querying.
>
>
> `Weakness 2: Training and evaluation are conducted on small dataset.`
>
> Thank you for this comment. We agree that the scale of the training and evaluation datasets (HyperNeRF and Neu3D) is indeed limited, and we have explicitly acknowledged this point in the Limitations section. That said, our primary contribution lies in **proposing a transformer-based feed-forward unified framework for 4D language grounding**, which is fundamentally different from previous per-scene optimization approaches such as 4D LangSplat. Even on these smaller-scale benchmarks, our method has successfully demonstrated the feasibility and advantages of a generalizable architecture, achieving competitive performance without scene-specific retraining.
>
> In future work, we plan to scale up the training data to further improve the model's generalization ability across more diverse and complex 4D scenes. We appreciate the reviewer's insightful feedback, which will help guide our next steps.
>
> `Question 2: Supplementary materials are not attached.`
>
> We thank the reviewer for bringing this to our attention. The supplementary materials, which include the video files, have now been submitted. We regret the omission in our original submission.

---

> ### Author Response · Authors · 2025-11-29
> **Rebuttal By Authors (Part 2/2)**
>
> `Question 3: Generalization across scenes or datasets.`
>
> Based on the reviewer's valuable feedback regarding the importance of evaluating model generalization, we have added two new experiments in the revised appendix: **cross-dataset generalization** and **cross-query generalization**.
>
> ## **Cross-dataset generalization**
> To assess cross-dataset generalization, we evaluated our model on the Objectron dataset [1], which was completely unseen during training. As shown in **Figure 7 of the revised paper**, although our model was trained with only a limited amount of data, it demonstrates encouraging generalization capability. This indicates that our pipeline possesses strong inherent robustness, and its performance can be further improved with larger training datasets.
>
> ## **Cross-query generalization**
>
> ##### *Table 1. Generalization experiments across different queries. We evaluated the performance of different time-sensitive queries on 4DLangSplat and 4DLangVGGT to explore their generalization capabilities across diverse queries.*
>
> | Query          |            |  **americano** |       |          |   **chick-chicken**   |      |
> |----------------|-----------|-------------------|--------|-----------|-------------------|--------|
> |                | Raw Query | Paraphrased Query | Δ      |  Raw Query | Paraphrased Query | Δ     |
> | 4DLangSplat    | 66.07     | 51.34             |-14.73 | 90.62     | 83.26             |-7.36  |
> | 4DLangVGGT     | 67.77     | 64.82             |-2.95  | 93.44     | 90.36             |-3.08  |
>
> We conducted a query-level generalization study, with results provided in Table 1. In this experiment, the original evaluation queries were replaced with semantically similar yet syntactically different expressions to test the model's robustness to linguistic variations. The results show that our model remains stable under such query changes and exhibits better cross-query generalization compared to 4DLangSplat. The modified queries used in this experiment are listed below:
>
> ### **Query for americano**
> - Raw query #1: Glasses contain light-colored liquid.
> - Raw query #2: Glasses contain dark brown liquid.
> - Paraphrased query #1: Glasses are filled with a light-colored liquid.
> - Paraphrased query #2: Glasses hold a deep brown-colored liquid.
>
> ### **Query for chick-chicken**
> - Raw query #1: Closed chicken container.
> - Raw query #2: Opened chicken container.
> - Paraphrased query #1: A chicken container that's sealed shut.
> - Paraphrased query #2: A container of chicken that is open.
>
>
> [1] Adel Ahmadyan, Liangkai Zhang, Artsiom Ablavatski, Jianing Wei, Matthias Grundmann, Objectron: A Large Scale Dataset of Object-Centric Videos in the Wild With Pose Annotations, Proceedings of the IEEE/CVF Conference on Computer Vision and Pattern Recognition (CVPR), 2021, pp. 7822-7831

---

### Meta-Review · Area_Chair_VbQK · 2025-12-04

**Summary:**

This paper proposes a Transformer-based, feed-forward unified framework for 4D language grounding. The core idea is to leverage the strong geometric priors of StreamVGGT and augment it with two heads to enhance semantic modeling. Experiments show improvements on both time-agnostic and time-sensitive benchmarks.

At the initial rating stage, one reviewer voted for acceptance, while the other two were negative.  Although the idea is good, the usage of LLM-based caption is questionable, and all the reviewers were concerned with the experiments, including: 1. generalization on larger datasets; 2. more ablation studies; 3. marginal per-scene gains; 4. missing efficiency analysis.

After the rebuttal, the authors partially address the above-mentioned concerns. However, the illustrations of 1. usage of LLM-based caption and 2. generalization on larger datasets are not convincing, failing to provide more experimental results and comparisons for justification. Moreover, the per-scene gains are also limited.

Based on the above reasons, the AC recommends Rejection. The authors are encouraged to carefully revise their work according to the comments of the reviewers.

**Reviewer Concerns:**

The authors have addressed:
1. Cross-dataset and cross-query evaluations.
2. More ablations of the DPT layer, frozen vs.\ fine-tuned encoder, and detailed analyses of time-agnostic vs.\ time-sensitive semantic supervision.
3. Efficiency analysis of training cost and inference speed.
4. More detailed clarifications and complements.

The main concerns remain:
1. Missing evaluation on large-scale datasets.
2. More discussion and usage of more effective temporal caption tools.
3. Marginal per-scene gains.

**Reviewer Scores:**

Since the concerns of missing evaluation on large-scale datasets, more discussion and usage of more effective temporal caption tools, and marginal per-scene gains are not addressed, the negative reviewers q5cc and 1e88 may not change their scores after discussion.

---

### Decision · Program_Chairs · 2026-01-26

Reject